# Evaluation of the reliability and validity of computerized tests of attention

Robert Langner[1,2], Frank Scharnowski[3,4,5,6], Silvio Ionta[7], Carlos E. G. Salmon[8], Brian J. Piper[9,10], Gustavo S. P. Pamplona[4,7,8,11] *

1 Institute of Systems Neuroscience, Medical Faculty, Heinrich Heine University Düsseldorf, Düsseldorf, Germany, 2 Institute of Neuroscience and Medicine, Brain & Behaviour (INM-7), Research Centre Jülich, Jülich, Germany, 3 Department of Cognition, Emotion, and Methods in Psychology, University of Vienna, Vienna, Austria, 4 Department of Psychiatry, Psychotherapy and Psychosomatics, Psychiatric Hospital, University of Zürich, Zürich, Switzerland, 5 Neuroscience Center Zurich, University of Zurich and Swiss Federal Institute of Technology, Zurich, Switzerland, 6 Zurich Center for Integrative Human Physiology (ZIHP), University of Zurich, Zurich, Switzerland, 7 Sensory-Motor Laboratory (SeMoLa), Jules-Gonin Eye Hospital/Fondation Asile des Aveugles, Department of Ophthalmology/University of Lausanne, Lausanne, Switzerland, 8 InBrain Lab, Department of Physics, University of São Paulo, Ribeirão Preto, Brazil, 9 Department of Medical Education, Geisinger Commonwealth School of Medicine, Scranton, PA, United States of America, 10 Center for Pharmacy Innovation and Outcomes, Forty Fort, Pennsylvania, United States of America, 11 Rehabilitation Engineering Laboratory (RELab), Department of Health Sciences and Technology, ETH Zurich, Zurich, Switzerland

* gsppamplona@gmail.com

**Data Availability Statement:** All data and analysis files are available on the public GitHub repository: https://github.com/gustavopamplona/PsychEval_attention.

## Abstract

Different aspects of attention can be assessed through psychological tests to identify stable individual or group differences as well as alterations after interventions. Aiming for a wide applicability of attentional assessments, Psychology Experiment Building Language (PEBL) is an open-source software system for designing and running computerized tasks that tax various attentional functions. Here, we evaluated the reliability and validity of computerized attention tasks as provided with the PEBL package: Continuous Performance Task (CPT), Switcher task, Psychomotor Vigilance Task (PVT), Mental Rotation task, and Attentional Network Test. For all tasks, we evaluated test-retest reliability using the intraclass correlation coefficient (ICC), as well as internal consistency through within-test correlations and split-half ICC. Across tasks, response time scores showed adequate reliability, whereas scores of performance accuracy, variability, and deterioration over time did not. Stability across application sites was observed for the CPT and Switcher task, but practice effects were observed for all tasks except the PVT. We substantiate convergent and discriminant validity for several task scores using between-task correlations and provide further evidence for construct validity via associations of task scores with attentional and motivational assessments. Taken together, our results provide necessary information to help design and interpret studies involving attention assessments.

**Funding:** F.S. was supported by the Swiss National Science Foundation (grants BSSG10_155915, 100014_178841, 32003B_166566, and PP00P1_170506/1), the Foundation for Research in Science and the Humanities at the University of Zurich (STWF-17-012), and the Baugarten Stiftung. S.I. was supported by the Swiss National Science Foundation (grants PZ00P1_170506/1 and PP00P1_202665/1). G.S.P.P was supported by the Brazilian National Council for Scientific and Technological Development (CNPq), the Brazilian National Council for the Improvement of Higher Education (CAPES) and the Swiss Government Excellence Scholarship. The funders had no role in study design, data collection and analysis, decision to publish, or preparation of the manuscript.

**Competing interests:** The authors have declared that no competing interests exist.

## 1. Introduction

Attention is a complex, multi-faceted construct. Far from being a unitary mental function, there are several aspects representing attention unified by the creation of a selective processing focus [1] supported by multiple brain networks [2–6]. Different facets of attention can be assessed via computerized attentional tests, which is important for characterizing individual attentional abilities that can be manipulated by interventions or affected by developmental changes as well as psychiatric or neurological conditions.

The assessment of attentional abilities in clinical practice, education, and research should rely on tests with sound psychometric properties. Without demonstrated reliability and validity, there can be no confidence in the interpretation of the measurements [7]. Reliability refers to the precision and, as a consequence, to the consistency and reproducibility of an assessment [8]. Reliability is an important psychometric requirement and can be evaluated at successive (test-retest reliability, usually measured with a test-retest intraclass correlation coefficient (ICC)) and at simultaneous events (e.g., split-half reliability, which can be measured with a split-half ICC) [9], and may also involve stability across populations and over time. Validity, in turn, describes to which degree an instrument truly measures the construct it purports to assess [8] and is often not established once but built up from gradual and cumulative evidence [10]. Validity may concern comparisons between the task content and definitions of a construct (based on hypothesis testing) and/or relationships (measured with linear correlations) with other related and unrelated scores obtained from a variety of instruments. Thus, the expected presence and absence of correlations with other instruments' scores contributes to validation.

The Psychology Experiment Building Language (PEBL) is an open-source system designed for creating and running computerized cognitive tasks to tax a large variety of cognitive processes or mental functions [11]. Among other cognitive functions, PEBL enables the assessment of various aspects of attention via a range of tasks. Although widely utilized [3], there is currently limited evidence on the psychometric properties of the available tasks in PEBL [3, 12–17]. Such studies, altogether, show only fragmented evidence of the psychometric properties across PEBL tasks dedicated to measuring attention. In fact, evidence on reliability or validity is entirely lacking for certain PEBL tasks (for example, the Switcher task), an evaluation across sites of application has never been done, and data on relationships between PEBL's attentional measures and cognitive questionnaires are scarce. Thus, evaluating the reliability and validity of PEBL tasks measuring aspects of attention is an important issue. This study will focus on the following five tasks: (i) The Continuous Performance Task (CPT) [12, 18, 19] is a widely used task that aims to measure the individual ability to sustain attention. It is a go/no-go task, in which targets and non-targets are randomly presented and require executing or withholding a speeded response, respectively. (ii) The Switcher task [20] was designed to determine one's ability to switch between different task sets (i.e., from one instruction to another), reflecting the individual level of cognitive flexibility in orienting or suppressing attention to distinct task rules. Switching costs measured by this task are held to represent an additional reconfiguration process or interference of cognitive processing from previous trials [21]. (iii) The Psychomotor Vigilance Task (PVT) [22–24] is a simple reaction time (RT) task that aims to measure the level of intrinsic alertness (i.e., readiness to respond to unexpected stimulus onsets) and its maintenance over time (i.e., vigilance). (iv) The Mental Rotation task [3, 25, 26] involves the transformation of visuo-spatial characteristics of an image without physically moving it (visual imagery) until it matches a target image. Performance is assessed via measuring response latency, which is proportional to the rotation angle, reflecting greater task difficulty with increasing angles. (v) The Attentional Network Test (ANT) [27] is designed to

provide separate measurements for three facets of attention: phasic alerting, by comparing responses to cued versus uncued stimuli; endogenous spatial orienting, by comparing responses to directional- versus center-cued stimuli; and conflict resolution, by comparing responses to congruent versus incongruent stimuli.

This study aimed to evaluate the reliability and validity of these five tasks as implemented in the PEBL software package. This evaluation was achieved through (i) assessments of task-condition-specific effects, to evaluate the consistency of task-induced affects with previous literature; (ii) examination of test-retest reliability and internal consistency, using test-retest and split-half intraclass correlation coefficients, respectively; (iii) sensitivity to practice, by testing for differences in task scores between days of application; (iv) stability of application across sites, by testing for differences in task scores across sites of application; (v) evaluation of reliability/internal consistency of a task using within-task correlations, as well as convergent/discriminant validity using between-task correlations; (vi) and evaluation of construct validity through associations of task scores with questionnaire scores related to attentional and motivational aspects. Using three datasets collected in two different countries, we evaluated reliability and validity in PEBL tasks' performance measures: performance speed and accuracy, performance deterioration with time, within-occasion performance variability, and other specific measures.

## 2. Methods

### 2.1. Participants

Fifty-two young adults participated in three different Studies (Table 1). Studies I and II were performed in Switzerland (University of Zurich, Zurich), and Study III was performed in Brazil (University of Sao Paulo, Ribeirao Preto). They were part of neurofeedback investigations based on functional magnetic resonance imaging (fMRI) [28]. Here, we focus on aspects of the behavioral assessment previously unpublished. In order to avoid potential influences of the experimental intervention on the behavioral data, we here only considered the first measurement day for Studies I and III. Two days of assessment application, separated by one week (mean = 7.2 ± 0.4 days, range [7, 8]), were considered for Study II, as the test-retest analyses were only performed for this group with no intervention between the two assessments.

All participants committed to abstain from alcohol or psychoactive drugs in the days of experiment as well as on the previous day; they were asked to maintain their regular caffeine morning routine, but not to ingest it directly right before the experiment. For Study II, participants committed to maintain the same bedtime and wake-up time for both days of assessment and to restrict alcohol ingestion to a moderate level between assessments. Only right-handed individuals were included (score ≥ 60 in the Edinburgh Handedness Inventory [29]. For Studies I and II, only individuals with sufficient English language skills were recruited. None of the recruited participants had a self-report history of severe psychological or neurological disorders associated with attentional abilities. All participants had normal or corrected-to-normal vision.

**Table 1. Characteristics of the three samples.**

| Study | n (n females) | Age mean (SD) | Age range |
|---|---|---|---|
| Study I | 17 (6) | 27.7 (3.2) | 22.6, 34.5 |
| Study II | 15 (5) | 25.9 (4.1) | 20.6, 35.6 |
| Study III | 20 (11) | 24.7 (3.4) | 20.0, 31.7 |
| Total | 52 (22) | 26.0 (3.7) | 20.0, 34.5 |

Participants in Studies I and II were compensated by CHF 25 per hour at the end of the experiment; participants in Study III did not receive monetary compensation. Participants in Brazil provided consent approved by the Research Ethics Committee of University of São Paulo and participants in Switzerland provided consent approved by the local ethics committee of the Canton of Zurich in Switzerland. All participants provided written informed consent before entering the study. All procedures were in agreement with the Declaration of Helsinki.

## 2.2. Experimental procedures and measures

Tasks for all Studies were performed using desktop computers (one computer for Studies I and II and another one for Study III), with keyboards and mice externally built and equipped with Microsoft Windows. The PEBL tasks [11] (accessible at http://pebl.sourceforge.net/download.html)), as described in detail below, were previously chained in the order given below and run systematically for each subject. Computers were not connected to the internet and on-screen notifications were disabled. Task execution reported for Studies I and II was conducted on an AMD A8-7410 2.2 GHz, 4 Gb RAM, Windows 10, monitor 23.8", and for Study III on an Intel® Pentium® IV 2.8 GHz, 1 Gb RAM, Windows 8, monitor 21".

Participants were instructed to sit comfortably in front of the computer screen in dedicated and silent rooms. For shielding against external noise, participants received memory foam earbuds in Studies I and II, whereas in Study III the room was isolated from sound entering from the surroundings. The testing rooms had lights on for Studies I and II, whereas lights were switched completely off for Study III. During the testing, participants were alone in the room. Before testing, we asked participants whether they wanted to use the restroom or if they needed anything else prior to the testing. All participants committed to leaving their mobile phones outside the application room.

Participants received written instructions for Studies I and II and oral instructions plus a slide presentation about the tasks in Study III (Fig 1A). They could ask questions prior to the testing session to the experimenters and both written instruction and slide presentation contained screenshots from the tasks. Directly before each task, instructions were once again presented on-screen to the participants, in English for Studies I and II and in Brazilian-Portuguese for Study III, as embedded in the PEBL implementation of each task. In general, all instructions contained information about task duration, to which stimuli to respond or not, whether the responses should be given through the keyboard or the mouse and which key to use, and information about different conditions. Additionally, all participants were asked to respond to the stimuli as accurately and as fast as possible. Five-minute breaks were included between the second and the third tasks for all Studies, and between the fourth and the fifth tasks for Studies I and II (Study III only comprised four tasks) (Fig 1A). Tasks were therefore presented in groups of two, and participants were instructed to leave the testing room during the breaks. Questionnaires were administered in a different room, prior to the testing session (Fig 1A). The total duration of the application, including breaks, was 1h25min ± 9min for Study I, 1h21min ± 3min for the first day of Study II, 1h18min ± 5 min for the second day of Study II, and 56min ± 6min for Study III (the latter study had one task less and one break less, as compared to Studies I and II).

**Studies I and II comprised a battery of five tasks.** CPT, Switcher, PVT, Mental Rotation, and ANT; Study III consisted of four tasks: CPT, Switcher, PVT, and Stroop (Fig 1A). Therefore, CPT, Switcher, and PVT were common to all studies (and applied in the same order), but Studies I and II included Mental Rotation and ANT and Study III included the Stroop task. As our evaluation of reliability relies on repeated assessments and stability across sites, and the Stroop Task was applied at only one site in a single application, this task was not further

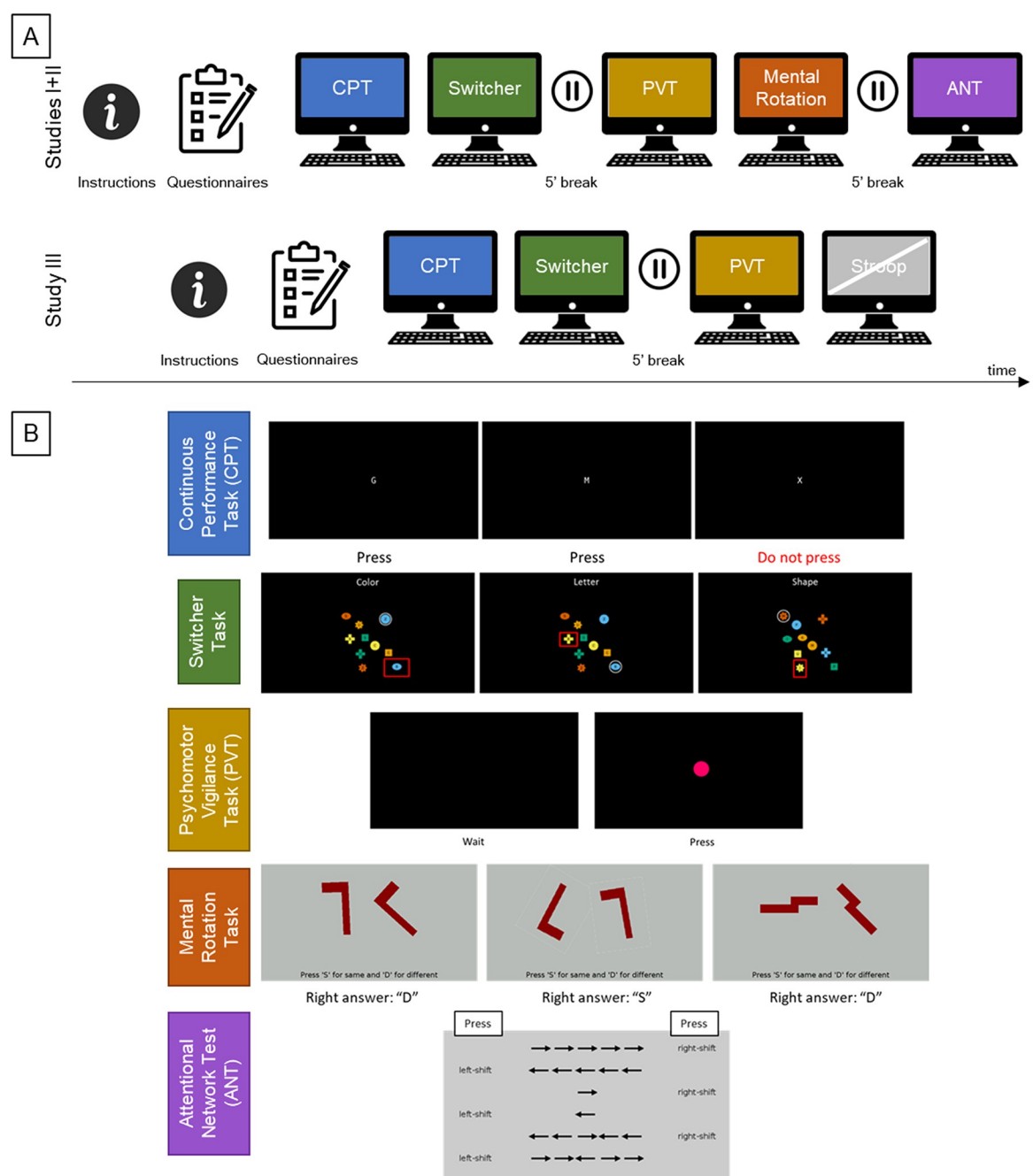

**Fig 1.** (A) Timeline of the studies. Upon receiving the instructions, participants completed the questionnaires (the Dundee Stress State Questionnaire (DSSQ) and the Cognitive Failures Questionnaire (CFQ) for Study I, only the DSSQ for Study II, and only the Brunel Mood Scale (BRUMS) validated to Brazilian-Portuguese for Study III). Next, participants performed the computerized tests, always in the same order (Continuous Performance Task (CPT), Switcher Task, Psychomotor Vigilance Task (PVT), Mental Rotation, and Attentional Network Test (ANT) for Studies I and II, and CPT, Switcher, PVT, and Stroop Task for Study III. The results for the Stroop Task were not analyzed in this study. Participants took five-minute breaks after conducting two tasks. (B) Screenshots of representative conditions and instructions for the PEBL attention tasks used. Tasks are color-coded as in A.

considered here. Furthermore, since Mental Rotation and ANT were applied at only one site (Studies I and II), these tasks were not considered for the assessment of stability across sites. Besides the computerized tasks, we also applied some questionnaires for psychometric

assessments: the Dundee Stress State Questionnaire (DSSQ) [30] and the Cognitive Failures Questionnaire (CFQ) [31] for Study I, only the DSSQ for Study II, and only the Brunel Mood Scale (BRUMS) validated to Brazilian-Portuguese [32] for Study III. All the questionnaires have been reported to have acceptable to very good reliability: Cronbach's alpha values were found to be in the range of 0.76–0.89 for DSSQ [30], 0.92–0.93 for CFQ [33], and 0.76–0.85 for BRUMS [34]. In Study I, some data from individual participants was missing due to technical issues: from one participant for CPT, Switcher task, and CFQ; from two participants for Mental Rotation task, ANT, and DSSQ. In Study III, three participants did not respond to the BRUMS questionnaire.

In the following section we describe the design of the attention tasks administered as embedded in the PEBL software, as well the measures extracted from each task (Fig 1B).

**Continuous Performance Task (CPT).**   This task is similar to the one reported by [18]. Subjects were requested to respond by pressing the space key of the keyboard to random white letters on a black background, as soon as they appeared on screen (go stimuli). However, subjects were also requested to withhold their response to the letter "X" (no-go stimuli), which constituted 10% of the trials. In total, 360 letters were presented, and this task lasted for approximately 14:30 min. The inter-stimulus intervals (ISIs) were 1, 2, or 4 s, randomly varied between blocks of 20 trials each, with each ISI length being employed in six blocks. Stimuli were shown for 250 ms and terminated for responses faster than that. RT and accuracy were recorded for every trial.

**Switcher task.**   This task was initially reported in [35]. In its PEBL implementation as used here, participants were requested, according to a switching feature rule, to identify pairs of figure elements among ten figures (which were unique combinations of five colors, five shapes, and five letters) displayed on a black screen. Each figure matched another one in only one feature (color, shape, or letter). In each trial, one figure was highlighted with a white outline and the subject was requested to click on the matching figure according to a current feature rule. The feature rules were shown on top of the screen and updated every trial between shape, color, and letter. A new feature rule was then given directly after a response to the correct figure, and the subject had to switch attention and click on the matching figure according to the new relevant feature rule. After a short practice round, each participant was presented with nine blocks, each of which performed until twelve correct answers were recorded. In the first three blocks, participants switched between two feature rules, each of the possible combination pairs per block (type 1: condition alternate switch). In the three subsequent blocks, participants switched between the three feature rules in a consistent order that changed for every block (type 2: condition fixed switch). In the last three blocks, participants switched between the three feature rules in a random order (type 3: condition random switch), in a way that the next rule could not be anticipated. The task lasted for approximately five minutes. RT and accuracy were recorded for every trial.

**Psychomotor Vigilance Task (PVT).**   This task is similar to the one reported by [22]. Subjects were required to respond by pressing the space key of the keyboard as soon as a circle appeared on a black screen. The circle disappeared as soon as the response was given, and the recorded RT was fed back on the screen. The circle was red (higher contrast) for Studies I and II, and blue (lower contrast) for Study III. Inter-stimulus intervals randomly varied between 2 and 12 s. In total, 121 trials were presented, and the task lasted for approximately 17 min. RT was recorded for every trial.

**Mental rotation task.**   This task is a bidimensional version of the classical test introduced by [26]. Subjects were requested to respond with pressing either "D" or "S" keys of the keyboard to indicate whether or not a pair of red figures presented on a gray screen was mirrored on the plane of the screen, respectively (condition Mirror). Figures were rotated with respect

to each other in angles of 0˚, 45˚, 90˚, 135˚, or 180˚ (condition Angle). Pairs of two different figures were shown, namely L-shape (defined as the familiar figure) and bolt (the unfamiliar figure) (condition Figure). In total, 128 trials were presented, and the task lasted for approximately 4:30 min. RT and accuracy were recorded for every trial.

**Attentional Network Test (ANT).** This task is based on the one introduced by [27]. Participants were requested to respond to a left- or right-pointing arrow (target) by pressing either the left or right shift key of the keyboard, respectively. The arrows were black and shown on a gray screen. The target appeared either above or below a steady fixation cross in the middle of the screen and disappeared as soon as the response was given. The target was embedded in-between arrows of the same or different direction, or not accompanied by other arrows at all, defined as congruent, incongruent, and neutral conditions, respectively (condition Congruence). In addition, visual spatial cues for exogenous attentional orienting were presented 100 ms before target onset (condition Cue). Cueing conditions comprised: center cued (asterisk in the middle of the screen), top-bottom cued (one asterisk at the top and one at the bottom, at the positions where the arrow might or not appear), direction cued (asterisk where the target will appear), and uncued (no asterisk). In total, 312 trials were shown (the first 24 for practice purposes only), and the task lasted for approximately 22 min. RT and accuracy were recorded for every trial.

**Self-report questionnaires.** To evaluate to which extent attention-related individual characteristics subjectively perceived are associated with the various PEBL attention task scores (construct validity), we also analyzed relationships between task performance and scores obtained from three self-report questionnaires. (1) The DSSQ [30] captures information about the perceived attentiveness, motivation, and stress of participants in performance situations. This questionnaire is sensitive to state fluctuations over time, reflecting attentional changes with time on task or due to an intervention. It is comprised of 4- and 5-point Likert-type rating scale questions and yields several subscores: energetic arousal, tense arousal, hedonic tone, anger/frustration, success motivation, intrinsic motivation, self-focused attention, self-esteem, concentration, control and confidence, task-related interference, and task-irrelevant interference. (2) The CFQ [31] assesses the frequency with which one experiences cognitive failures or absent-mindedness, which is a stable inverse measure of attentional characteristics (or "traits") with respect to everyday life. It consists of 5-point Likert-type rating scale questions and yields a single score. (3) The BRUMS [36] (validated Brazilian-Portuguese language version by [34]) evaluates subjective mood states perceived by the participants. It consists of 5-point Likert-type rating scale questions and yields six subscores: tension, depression, anger, vigor, fatigue, and confusion.

## 2.3. Data analysis

Each PEBL task application resulted in a logfile from which the information of interest was extracted and organized according to Study, day, participant, and condition. We computed the task scores for each condition of interest using MATLAB (The MathWorks, Natick, MA, USA) and statistical evaluation was performed through RStudio (https://rstudio.com/). The task scores were: RT, rate of commission errors, rate of omission errors, standard deviation of the RT, and slope of RT for CPT; RT, rate of errors, and standard deviation of the RT for Switcher task; RT, rate of lapses, rate of premature responses and standard deviation of the RT for PVT; RT and accuracy for Mental Rotation task; RT and accuracy for ANT (Table 2). We evaluated each task according to its availability. For each task, within-subject RT outliers over trials were excluded by iteratively removing data points outside the range average +/- three standard-deviations. CPT data from one individual in Study III was not used in any analysis

**Table 2. Conditions of interest and measurements for each task.**

| Task | Conditions | Measurements |
|---|---|---|
| CPT | ISI | • Reaction time (average across correct trials)<br>• Standard deviation (across correct trials)<br>• Commission errors (proportion of go responses in no-go trials)<br>• Omission errors (proportion of no-go responses in go trials) |
| | Total | • Slope of RT (of a linear regression fitted on the values of six average reaction times within equally divided blocks over time) |
| Switcher | Type | • Reaction time (average across correct trials)<br>• Standard deviation (across correct trials)<br>• Errors (proportion of wrong responses) |
| PVT | Total | • Reaction time (average across trials, excluding early responses and first trial)<br>• Standard deviation (across trials, excluding early responses and first trial)<br>• Lapses (number of responses longer than 500 ms, excluding first trial)<br>• Premature responses (responses before the stimulus onset, excluding first trial)<br>• Slope of RT (computed similar to CPT, first trial removed) |
| Mental Rotation | Angle, Figure, and Mirror | • Reaction time (average across trials, excluding early responses)<br>• Accuracy (proportion of correct responses) |
| ANT | Cue and Coherence | • Reaction time (average across trials, excluding early responses)<br>• Accuracy (proportion of correct responses) |
| | Total | • Slope of RT (computed similar to CPT)<br>• Alerting (average reaction times across uncued minus average reaction time across top-down cued stimuli)<br>• Orienting (average reaction time across center-cued stimuli minus average reaction time across directional-cued stimuli)<br>• Conflict (average reaction time across incongruent stimuli minus average reaction time from congruent stimuli) |

because of rate of omission errors higher than the between-subject average plus four standard deviations. Because of the typically skewed distribution of the RT, we computed the logarithmic transformation and the median (instead of the average) of the RT for each condition for all tasks except ANT, for which we computed its individual average for each condition, following the standard procedures reported in [27]. Practice trials (for Switcher Task and ANT) were not considered in any analysis.

**Assessments of task-condition-specific effects.**   We first examined the consistency of the results according to previous literature by examining group-level differences between task conditions. For this assessment, only data from a single application were used and, therefore, only the first day of application for Study II was considered. Prior to analysis, we removed individual RT measures identified as extreme outliers (more than three times the interquartile range below the first quartile or above the third quartile) across conditions (ISI in CPT, Type in Switcher, Angle in Mental Rotation, and Congruence in ANT) (library 'rstatix'). For the evaluation of RT across conditions, further individual data was removed upon violation of the normality assumption according to the Shapiro-Wilk test. Continuous variables (RT, standard deviation) were analyzed with N-way mixed ANOVA. Within factors were ISI for CPT; Type for Switcher task; Mirror, Figure, and Angle for Mental Rotation task; and Congruence and Cue for ANT. Site was defined as the between factor (for ANT, Switcher Task, and PVT). Discrete variables (errors, accuracy) were evaluated with generalized linear mixed-effects models (library 'lme4') and conditions were designed as nested variables within subjects when the number of condition levels was higher than two (for the estimation of variance). For performance differences between two conditions (slopes, PVT measures), parametric tests were applied whenever the assumptions of normality were fulfilled (using the Shapiro-Wilk normality test); otherwise, non-parametric tests were used. We computed post-hoc analyses for

significant main effect or interaction following ANOVA or generalized linear mixed-effects model (library 'emmeans'), and p-values were adjusted for multiple comparisons by the Tukey method. As estimated of effect sizes, we computed the partial eta-squared (library 'DescTools') for main effects and interactions and Cohen's d for pairwise comparisons.

**Evaluation of test–retest reliability and internal consistency.** We investigated the test-retest reliability and internal consistency of the task by computing test-retest and split-half intraclass correlation coefficients (ICCs), respectively. We computed the ICC of each score and the boundaries of a confidence interval of 95% (library 'rel'). The split-half ICC was obtained by grouping the dataset in odd and even trials and computing the ICC between them. The ICC reflects reliability by describing how well measures in the same group resemble each other, through the ratio of true variance over true variance plus error variance. According to Koo & Li (2016), a two-way mixed-effects with absolute agreement and single measurement (terminology based on [37]) is the recommended approach for the assessment of test-retest (and split-half) reliability. RT and accuracy were averaged across Angle conditions in the Mental Rotation task and averaged across the Coherence condition in the ANT. The ICC reflects the proportion of variance between groups, therefore, negative values in the lower confidence boundary were replaced by zero in order to remain meaningful [38]. The ICC is a more satisfactory measure of test-retest reliability than the Pearson correlation, as it reflects not only the correlation across measurements but also the degree of agreement between them [7].

**Evaluation of temporal stability.** We examined the sensitivity to practice of a task score by testing for differences between days of application. These comparisons of application days were made only for Study II (the only one with a double application). Despite the within factors described above (subsection "Assessments of task-condition-specific effects"), the within factor Day was also included in this evaluation. Outlier removal, described in the subsection "Assessments of task-condition-specific effects", was also applied for this evaluation. We computed N-way repeated-measures ANOVA for continuous variables and generalized linear mixed-effects models for discrete variables. Following Shapiro-Wilk testing, paired t-tests were computed for parametric distributions (RT in PVT; alerting, orienting, conflict, and RT slope in ANT), and dependent-group Wilcoxon signed-rank tests were computed for nonparametric distributions (slope of RT in CPT; lapses, premature responses, and slope of RT in PVT). We computed post-hoc analyses for significant main effects and interactions and estimated the effect sizes.

**Evaluation of stability across application sites.** We examined the stability across sites of application of a task score by testing for differences between sites. Comparisons between sites of application were made between Studies I+II and Study III considering the analysis described in "Assessments of task-condition-specific effects" and the interactions and main effects containing the between factor Day. Only the CPT and the Switcher task were evaluated for stability across application sites because they were the only two tests performed in different countries (the PVT presented substantial design differences and experimental effects were evaluated instead).

**Within- and between-task correlations.** We assessed the reliability/internal consistency of a task score through within-task correlations and the convergent/discriminant validity through between-task correlations. Here, internal consistency indicates how strongly different scores of a given task are related to each other and whether they represent the same (core) construct [8], quantifying within-occasion reliability. If scores are measuring a single construct, they should lead to more homogenous results and therefore higher internal consistency; conversely, the scores might be measuring more than one construct [8]. Additionally, variables thought to reflect similar constructs would be expected to be rather closely correlated to each other, indicating convergent validity; in contrast, measures reflecting unrelated constructs

should not correlate with each other, revealing discriminant validity [10]. In other words, construct validity is supported when correlations between different task scores are high for the same (or a similar) trait but low for different traits. We computed Spearman rank correlation coefficients between task scores across individuals, including all Studies. Only scores related to RT, slope of RT and ANT-specific measures were considered for this analysis–we did not included accuracy- or variability-related measures in this analysis because of the lack of test-retest availability described in the section 3.2.; however, slopes of RT were included in this analysis because of its relevance for evaluating performance over time. P-values were adjusted for multiple comparisons by the False Discovery Rate (FDR) at the level of each score.

**Associations with questionnaire scores.** To provide further evidence for construct validity, we computed the associations of task scores with questionnaire scores related to attentional and motivational aspects. We computed the twelve aforementioned DSSQ sub-scores, obtained from linear combinations of specific answers, a single CFQ measure by averaging all the answers provided, and the following six BRUMS sub-scores by averaging specific answers. Non-parametric Spearman rank correlation coefficients were then computed for each combination of questionnaire and task scores. Following the questionnaires applied in each study, we evaluated the associations between DSSQ and CFQ scores with the task measures from Studies I + II, and the associations between BRUMS scores and the task measures from Study III. As described in the previous paragraph, "Within- and between-task correlations", only scores related to RT, slope of RT and ANT-specific measures were considered for this analysis. P-values were FDR-adjusted for multiple comparisons at the level of scores per questionnaire.

## 3. Results

### 3.1. Assessments of task-condition-specific effects

For the CPT, we observed an increase in RT with longer ISIs (main effect of ISI, S1A Fig, S1 Table). We also observed a decrease in within-subject standard deviation with longer ISIs (main effect of ISI, S1C Fig, S1 Table). No effects of ISI on either Commission or Omission Errors were observed. Descriptive statistics about RT, commission errors, omission errors, and slope for the CPT across ISI are reported in S3 Table.

For the Switcher task, we observed no effect of switching condition on RT or accuracy. However, when only the second day of application is considered, RT was higher in the random than the alternate switch condition (main effect of Type, S2A Fig, S2 Table), indicating that the expected effects are detected only after some practice. We also observed that within-subject standard deviation increased with increasing switch difficulty (main effect of Type, S2B Fig, S1 Table). Descriptive statistics about RT and errors for the Switcher task across switching conditions are reported in S4 Table.

For the PVT, the lower visual stimulus contrast in the Brazilian dataset resulted in longer RT, higher number of lapses, fewer premature responses (S8 Table), but we observed no differences in RT slope (i.e., RT change with time on task). Descriptive statistics about RT, lapses, premature responses, and slope for the PVT are reported in S5 Table.

For the Mental Rotation task (S1 Table), we observed that RT increased with growing rotation angle (main effect of Angle, S3A Fig); with unfamiliar compared to familiar figures, mainly for larger angles (interaction Figure x Angle, S3B Fig); and with mirrored relative to non-mirrored figures, mainly for smaller angles (interaction Mirror x Angle, S3C Fig). The three-way interaction Mirror x Figure x Angle was also significant. We also found that accuracy decreased with larger rotation angle (main effect of Angle, S3D Fig). Descriptive statistics about RT and accuracy for the Mental Rotation task across rotation angle, figures, and same/different conditions are reported in S6 Table.

For the ANT (S1 Table), we observed that RT increased with spatially non-informative (vs. spatially informative) cues (main effect of Cue, S4A Fig); and with incongruent (vs. congruent) flanking arrows (main effect of Congruence, S4B Fig). Descriptive statistics about RT and accuracy for the ANT across congruence and cue, as well as alerting, orienting, conflict, and slope, are reported in S7 Table.

## 3.2. Evaluation of test-retest reliability and internal consistency

Test-retest ICC values for all tasks are reported in Fig 2. In general, RT shows good to excellent ICC, except for Switcher, CPT scores, and Mental Rotation task for the angles of 0˚ and 180˚. Furthermore, scores related to accuracy, variability in performance speed, and speed detriment with time on task exhibit poor to moderate ICC values, with only few exceptions. ANT alerting, orienting, and conflict presented moderate ICC values. Split-half ICC values for all tasks are reported in the S5 Fig, which shows acceptable internal consistency mainly for RT but mostly not for other measures.

## 3.3. Temporal stability

Practice effects were present for RT in all tasks (main effect of Day, all ps < 0.005) but the PVT (Fig 3, Table 3). Some practice effects across task conditions were observed for stimulus

**CPT**

| Condition | Reaction time | | | | Commission errors | | | | Omission errors | | | | Standard deviation | | | | Slope |
|---|---|---|---|---|---|---|---|---|---|---|---|---|---|---|---|---|---|
| | ISI1 | ISI2 | ISI4 | Total | ISI1 | ISI2 | ISI4 | Total | ISI1 | ISI2 | ISI4 | Total | ISI1 | ISI2 | ISI4 | Total | Total |
| Lower CB | 0.02 | 0.06 | 0.23 | 0.15 | 0 | 0 | 0 | 0 | 0.23 | 0 | 0 | 0 | 0 | 0.45 | 0 | 0.25 | 0 |
| IIC | 0.68 | 0.65 | 0.65 | 0.66 | 0.24 | 0.27 | 0.45 | 0.48 | 0.66 | 0 | 0.30 | 0.41 | 0.40 | 0.79 | 0.46 | 0.65 | 0 |
| Upper CB | 0.90 | 0.88 | 0.87 | 0.88 | 0.65 | 0.68 | 0.78 | 0.79 | 0.87 | 0.44 | 0.69 | 0.75 | 0.75 | 0.93 | 0.77 | 0.87 | 0.15 |

**Switcher**

| Condition | Reaction time | | | Error rate | | | Standard deviation | | |
|---|---|---|---|---|---|---|---|---|---|
| | Type1 | Type2 | Type3 | Type1 | Type2 | Type3 | Type1 | Type2 | Type3 |
| Lower CB | 0 | 0.03 | 0.11 | 0 | 0.36 | 0.24 | 0.32 | 0 | 0.29 |
| IIC | 0.51 | 0.58 | 0.56 | 0.10 | 0.73 | 0.65 | 0.71 | 0.31 | 0.73 |
| Upper CB | 0.84 | 0.85 | 0.82 | 0.57 | 0.90 | 0.87 | 0.89 | 0.70 | 0.90 |

**PVT**

| Condition | Reaction time | Lapses | Premature response | Slope |
|---|---|---|---|---|
| Lower CB | 0.76 | 0.31 | 0.27 | 0 |
| IIC | 0.91 | 0.70 | 0.67 | 0.36 |
| Upper CB | 0.97 | 0.89 | 0.88 | 0.73 |

**Rotation**

| Condition | Reaction time | | | | | Accuracy | | | | |
|---|---|---|---|---|---|---|---|---|---|---|
| | 0° | 45° | 90° | 135° | 180° | 0° | 45° | 90° | 135° | 180° |
| Lower CB | 0 | 0.49 | 0.65 | 0.55 | 0.07 | 0 | 0 | 0.13 | 0.07 | 0.11 |
| IIC | 0.47 | 0.80 | 0.87 | 0.82 | 0.55 | 0.47 | 0.33 | 0.59 | 0.54 | 0.58 |
| Upper CB | 0.78 | 0.93 | 0.95 | 0.94 | 0.82 | 0.79 | 0.71 | 0.84 | 0.82 | 0.83 |

**ANT**

| Condition | Reaction time | | | Accuracy | | | Alerting | Orienting | Conflict | Slope |
|---|---|---|---|---|---|---|---|---|---|---|
| | Congruent | Neutral | Incongruent | Congruent | Neutral | Incongruent | | | | Total |
| Lower CB | 0.57 | 0.64 | 0.58 | 0 | 0 | 0.23 | 0.32 | 0.07 | 0.13 | 0 |
| IIC | 0.84 | 0.86 | 0.84 | 0 | 0.48 | 0.65 | 0.70 | 0.59 | 0.59 | 0.13 |
| Upper CB | 0.94 | 0.95 | 0.94 | 0.27 | 0.79 | 0.87 | 0.89 | 0.82 | 0.84 | 0.59 |

■ Excellent  ■ Good  ■ Moderate  ■ Poor

**Fig 2. Intraclass Correlation Coefficients (ICC) of task scores for a one-week test-retest application with 15 participants.**
CPT = Continuous Performance Task; PVT = Psychomotor Vigilance Task; ANT = Attention Network Test; ISI1, ISI2, ISI4 = inter-stimulus interval of 1, 2, and 4 s, respectively; Type1, Type2, and Type3 = alternate switch, fixed switch, and random switch, respectively, CB = confidence boundary. Blue, green, yellow, and orange colors indicate excellent (ICC ≥ 0.9), good (0.75 ≤ ICC < 0.9), moderate (0.5 ≤ ICC < 0.75), and poor (ICC < 0.5) ICC values.

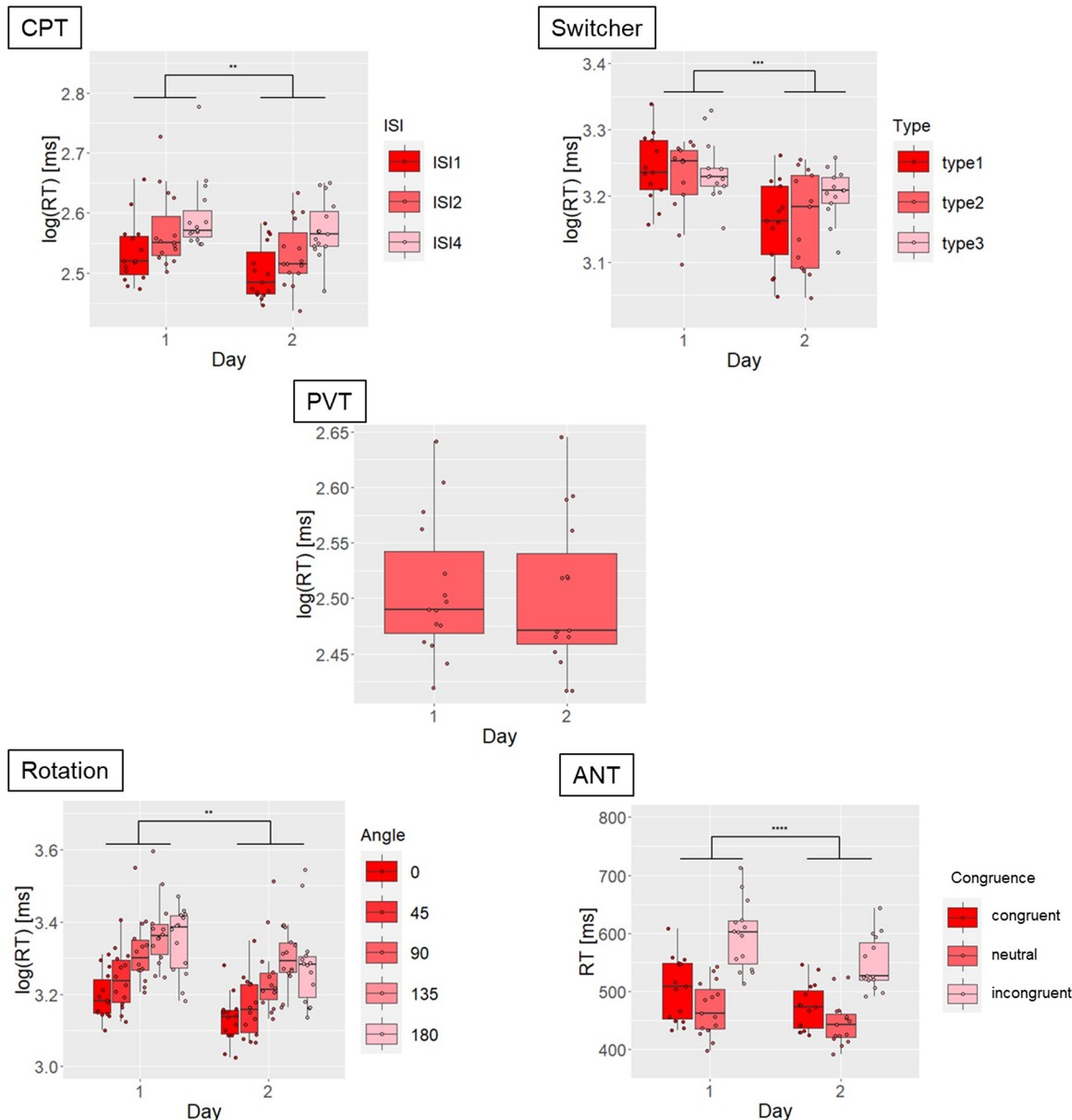

**Fig 3. Practice effects are present in reaction times (RT) for all applied tasks, except for PVT.** Asterisks represent significant differences (**** p < 0.0001, *** p < 0.001, ** p < 0.01). CPT = Continuous Performance Task; PVT = Psychomotor Vigilance Task; ANT = Attention Network Test.

congruence in ANT (i.e., interaction Congruence x Day) and ANT conflict component (S6 Fig, Table 3). We did not observe practice effects in other scores (all ps > 0.05).

## 3.4. Stability across application sites

We did not observe effects of Site on RT (Fig 4) and accuracy for the CPT and Switcher task across application sites (main effects of Site and interaction Site x condition, all ps > 0.05). For the Switcher task, we observed that the individual standard deviation was higher for random and fixed switch conditions, relative to the alternate condition, only in the Brazilian sample (interaction Site x Type, S8 Table).

**Table 3. Statistics for significant main effects and interactions with the factor Day and subsequent post-hoc analyses.**

| CPT[a] | | | | | |
|---|---|---|---|---|---|
| Reaction time–Main effect Day–$F_{(1,14)} = 13.5$, $\eta^2 = 0.49$, $p = 0.0025$ | | | | | |
| Day | Estimate (ms) | DoF[b] | t-value | Cohen's d | p-value[c] |
| Day1—Day2 | 32.7 | 14 | 3.67 | 0.87 | 0.0025 |
| Switcher | | | | | |
| Reaction time–Main effect Day–$F_{(1,12)} = 32.2$, $\eta^2 = 0.73$, $p = 0.0001$ | | | | | |
| Day | Estimate (ms) | DoF | t-value | Cohen's d | p-value |
| Day1—Day2 | 224 | 12 | 5.67 | 1.03 | 0.0001 |
| PVT[d] | | | | | |
| Reaction time–difference across Days | | | | | |
| Day | Estimate (ms) | DoF | t-value | Cohen's d | p-value |
| Day1—Day2 | 3.78 | 14 | 0.45 | 0.20 | 0.4 |
| Mental Rotation | | | | | |
| Reaction time–Main effect Day–$F_{(1,13)} = 15.9$, $\eta^2 = 0.55$, $p = 0.0016$ | | | | | |
| Day | Estimate (ms) | DoF | t-value | Cohen's d | p-value |
| Day1—Day2 | 290 | 13 | 3.99 | 0.52 | 0.0015 |
| ANT[e] | | | | | |
| Reaction time–Main effect Day–$F_{(1,14)} = 29.5$, $\eta^2 = 0.68$, $p < 0.0001$ | | | | | |
| Day | Estimate (ms) | DoF | t-value | Cohen's d | p-value |
| Day1—Day2 | 35.1 | 14 | 5.43 | 1.04 | 0.0001 |

| Reaction time–Interaction Day x Congruence–$F_{(2,28)} = 14.3$, $\eta^2 = 0.50$, $p < 0.0001$ | | | | | |
|---|---|---|---|---|---|
| Day | Congruence | Estimate (ms) | DoF | t-value | Cohen's d | Adj. p-value |
| Day1 | con–inc[f] | -90.6 | 38.9 | -16.6 | 2.85 | <0.0001 |
| | con—neu | 35.5 | | 6.49 | 1.27 | <0.0001 |
| | inc—neu | 126.1 | | 23.1 | 3.15 | <0.0001 |
| Day2 | con—inc | -75.4 | | -13.8 | 2.76 | <0.0001 |
| | con—neu | 27.1 | | 5.0 | 1.46 | <0.0001 |
| | inc—neu | 102.5 | | 18.7 | 2.96 | <0.0001 |
| Day1—Day2 | con | 32.9 | 18.6 | 4.72 | 1.08 | 0.0002 |
| | neu | 24.5 | | 3.51 | 0.86 | 0.0024 |
| | inc | 48.1 | | 6.90 | 1.28 | <0.0001 |

| Conflict–difference across Days | | | | | |
|---|---|---|---|---|---|
| Day | Estimate | DoF | t-value | Cohen's d | p-value |
| Day1—Day2 | 15.22 | 14 | 3.61 | 0.93 | 0.0029 |

Note. Statistics for differences between Days for PVT and ANT conflict are also shown.

[a]CPT = Continuous Performance Task

[b]DoF = degrees-of-freedom

[c]Adj. = adjusted (for multiple comparisons)

[d]PVT = Psychomotor Vigilance Task

[e]ANT = Attention Network Test

[f]con = congruent; neu = neutral; inc = incongruent.

## 3.5. Within- and between-task correlations

Substantial within-task correlations between different task conditions or trial types, which attest to a task's internal consistency, were observed for all RT-related scores, as can be gleaned from the positive correlation coefficients obtained for assessments on the first application day, given in Fig 5 (green clusters close to the main diagonal). A similar result was found for the second application day of Study II (i.e., containing practice effects) (S7 Fig).

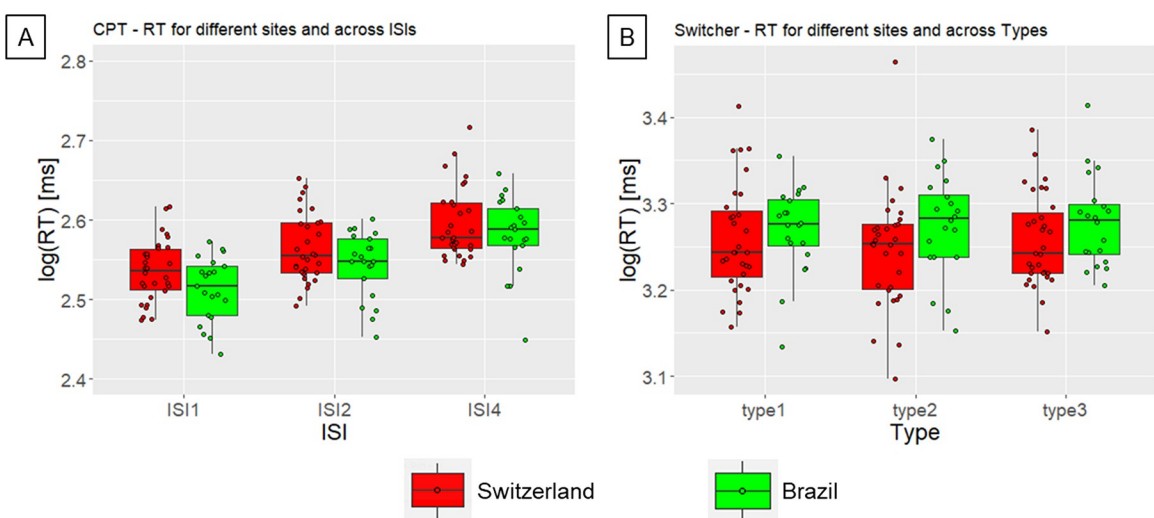

**Fig 4.** Both CPT (A) and Switcher (B) are reliable across application sites in terms of reaction times (RT). Red and green graphs represent RT for applications in Switzerland and Brazil, respectively. ISI1, ISI2, ISI4 = inter-stimulus interval of 1, 2, and 4 s, respectively. Type1, type2, type3 = alternate switch, fixed switch, random switch, respectively.

High between-task correlations, addressing the question of convergent validity, were observed for RT (i) between CPT and ANT and (ii) between Switcher and Mental Rotation tasks (Fig 5 and S8 Fig). Furthermore, moderate between-task correlations were observed (iii)

| | CPT RT ISI2 | CPT RT ISI4 | CPT slope | Switcher RT type1 | Switcher RT type2 | Switcher RT type3 | PVT RT | PVT slope | Rotation RT 0 | Rotation RT 45 | Rotation RT 90 | Rotation RT 135 | Rotation RT 180 | ANT RT congruent | ANT RT neutral | ANT RT incongruent | ANT alerting | ANT orienting | ANT conflict | ANT slope |
|---|---|---|---|---|---|---|---|---|---|---|---|---|---|---|---|---|---|---|---|---|
| CPT RT ISI1 | **0.86** | **0.83** | 0.31 | 0.37 | 0.27 | 0.15 | **0.39** | -0.21 | 0.33 | 0.18 | 0.17 | 0.38 | 0.32 | **0.61** | **0.61** | 0.48 | 0.13 | 0.14 | -0.05 | 0.01 |
| CPT RT ISI2 | | **0.92** | 0.28 | **0.38** | 0.10 | 0.12 | 0.32 | -0.25 | 0.32 | 0.15 | 0.16 | **0.46** | **0.40** | **0.59** | **0.62** | **0.52** | 0.16 | 0.16 | -0.02 | 0.02 |
| CPT RT ISI4 | | | 0.26 | **0.45** | 0.28 | 0.20 | 0.25 | -0.24 | 0.30 | 0.21 | 0.21 | **0.41** | **0.43** | **0.59** | **0.60** | **0.61** | 0.16 | 0.17 | 0.11 | 0.01 |
| CPT slope | | | | -0.05 | -0.03 | -0.22 | 0.12 | -0.05 | 0.22 | 0.06 | 0.08 | 0.29 | 0.15 | 0.30 | 0.19 | 0.20 | 0.02 | 0.20 | -0.18 | -0.18 |
| Switcher RT type1 | | | | | **0.66** | **0.72** | 0.20 | 0.07 | 0.36 | **0.43** | **0.51** | 0.32 | **0.50** | 0.27 | 0.28 | 0.31 | -0.10 | -0.38 | 0.19 | -0.04 |
| Switcher RT type2 | | | | | | **0.66** | 0.06 | -0.02 | 0.22 | **0.42** | **0.43** | **0.39** | **0.47** | 0.14 | 0.18 | 0.24 | -0.02 | -0.21 | 0.32 | 0.08 |
| Switcher RT type3 | | | | | | | 0.28 | 0.08 | 0.28 | **0.39** | **0.46** | 0.25 | **0.41** | 0.26 | 0.30 | 0.31 | 0.16 | **-0.56** | 0.29 | 0.04 |
| PVT RT | | | | | | | | 0.36 | 0.10 | 0.16 | 0.24 | 0.23 | 0.20 | 0.24 | 0.37 | 0.19 | **0.52** | -0.32 | 0.01 | -0.08 |
| PVT slope | | | | | | | | | -0.05 | 0.08 | 0.09 | -0.23 | -0.04 | -0.10 | 0.00 | -0.12 | **0.42** | **-0.57** | -0.08 | -0.16 |
| Rotation RT 0 | | | | | | | | | | **0.79** | **0.81** | **0.68** | **0.76** | **0.46** | **0.39** | **0.48** | -0.14 | -0.11 | 0.29 | -0.12 |
| Rotation RT 45 | | | | | | | | | | | **0.91** | **0.63** | **0.85** | 0.30 | 0.30 | 0.33 | 0.13 | -0.16 | 0.18 | -0.07 |
| Rotation RT 90 | | | | | | | | | | | | **0.63** | **0.87** | 0.27 | 0.30 | 0.28 | 0.08 | -0.22 | 0.16 | -0.06 |
| Rotation RT 135 | | | | | | | | | | | | | **0.83** | 0.24 | 0.24 | 0.33 | -0.07 | 0.01 | 0.28 | -0.06 |
| Rotation RT 180 | | | | | | | | | | | | | | 0.31 | 0.33 | 0.34 | 0.10 | -0.09 | 0.13 | 0.07 |
| ANT RT congruent | | | | | | | | | | | | | | | **0.95** | **0.89** | 0.18 | -0.13 | 0.21 | -0.02 |
| ANT RT neutral | | | | | | | | | | | | | | | | **0.83** | **0.40** | -0.17 | 0.17 | 0.04 |
| ANT RT incongruent | | | | | | | | | | | | | | | | | 0.04 | -0.10 | **0.57** | -0.10 |
| ANT alerting | | | | | | | | | | | | | | | | | | -0.24 | -0.29 | 0.18 |
| ANT orienting | | | | | | | | | | | | | | | | | | | -0.18 | -0.01 |
| ANT conflict | | | | | | | | | | | | | | | | | | | | -0.07 |

**Fig 5. Correlation among scores from the tasks on the first day of application.** Significant positive and negative correlations (FDR-corr. p < 0.05) are reported in green and red, respectively. Correlations with p < 0.05, uncorrected for multiple comparisons, are shown in bold. CPT = Continuous Performance Task; PVT = Psychomotor Vigilance Task; ANT = Attention Network Test; RT = reaction time; ISI1, ISI2, ISI4 = inter-stimulus interval of 1, 2, and 4 s, respectively; type1, type2, type3 = alternate switch, fixed switch, random switch, respectively.

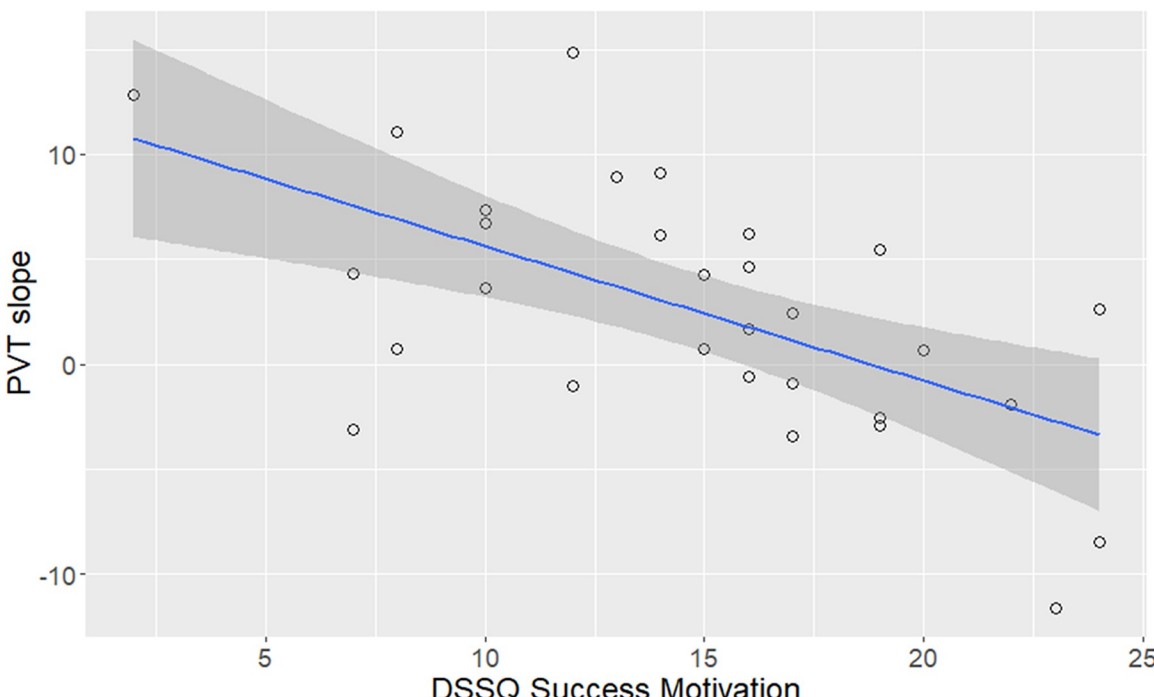

**Fig 6. Performance detriment is inversely proportional to motivation for success in the task (FDR-corr. $p < 0.05$).** Performance detriment was measured as the linear slope of reaction time (RT) through a Psychomotor Vigilance Task (PVT); success motivation was measured via the Dundee Stress State Questionnaire (DSSQ). The solid blue line and shaded area indicate the linear regression line and its confidence interval, respectively.

between PVT measures and ANT Alerting performance (only on the first day of application) and (iv) for RT between CPT and more difficult conditions (highly rotated stimuli) in the Mental Rotation task. Conversely, (v) ANT Orienting performance was anticorrelated to the PVT's RT slope and to the Switcher measures.

### 3.6. Associations with questionnaire scores

We found a significant relationship between PVT slope of RT (i.e., response slowing with time on task) and DSSQ success motivation (FDR-corrected $p < 0.05$) (Fig 6, S9 Table), indicating that perceived motivation to perform well during the experiment is correlated with keeping up intrinsic alertness. Task scores were also moderately correlated with other DSSQ scores (uncorrected $p < 0.05$), as well as highly correlated with CFQ (S8 Fig, S9 Table). We did not observe significant correlations between BRUMS and task scores.

### 4. Discussion

This study aimed to evaluate psychometric properties of several well-established tasks that tax different aspects of attention. Overall, the multi-faceted evidence on the tasks' reliability and validity reported here provides information that can guide other studies, in both basic and applied research, oriented toward the assessment of interindividual differences. All tasks evaluated here are included in the PEBL software package [11], which provides an accessible and open-source solution for researchers aiming to implement computerized tasks in their assessment procedures. Once the psychometric quality of specific PEBL tasks has been demonstrated, they could be an alternative to commercial tests, contributing to the democratization of science [3]. However, most of the available PEBL experimental paradigms still lack proper

evaluation of their reliability and validity. Here, we evaluated five computerized tasks written in PEBL tapping into various aspects of attention: CPT, Switcher task, PVT, Mental Rotation task, and ANT. For this, we examined parameters of test-retest reliability, internal consistency, stability over time and across application sites, convergent/discriminant validity, and construct validity based on attentional and motivational aspects.

## 4.1. Assessment of task-condition-specific effects

Before discussing the psychometric properties under scrutiny here, we examined the consistency of the group-level task effects with previous literature. The dependence of performance scores on task-specific conditions might also provide information in terms of construct validity [8].

For the CPT, we observed that the inter-stimulus interval affects RT (S1A and S1B Fig, S1 Table). Namely, RT increased with longer ISIs, as reported in [18]. The standard deviation of RT within each subject, reflecting the variability of this measurement, decreased with longer intervals (S1C Fig, S1 Table). We did not observe dependencies of commission and omission errors on ISI length in this task. This might be because errors in the CPT (with the parameters used here) are relatively rare events and any dependence on ISI length may be only observable in large samples [18].

Interestingly, for the Switcher task, we observed no dependence of RT or error rate on the type of switch condition (Figs 3 and 4), in discordance with [35]. In particular, longer response latencies would be expected for the random switch condition because of additional reconfiguration processes and/or stronger interference from preceding trials under conditions of unpredictable rule switching [21]. Therefore, despite its apparent face validity [8], the application of this task for evaluating the ability of switching attention is not recommended under the default parameters currently defined in PEBL. We will further discuss the validity of the Switcher task and propose an alternative application that may lead to appropriate psychometric properties in the section "Temporal stability" below.

The PVT features only one condition, requiring continued speeded stimulus detection (rather than discrimination). RT is recorded over time for responses to stimuli occurring at unpredictable times (i.e., variable ISIs), in order to assess sustained (intrinsic) alertness [39]. As expected, for low-contrast stimuli, we observed longer RT, more attentional lapses, and fewer premature responses (S8 Table). However, performance change with time on task (i.e., slope of RT) was not different between stimulus contrast levels, indicating a similar stability of alertness over time despite differences in perceptual difficulty. In contradistinction, lower stimulus contrast led to a higher number of attentional lapses compared to higher stimulus contrast.

For the Mental Rotation task, we observed the commonly reported pattern of performance varying with angle, i.e., longer RT and lower accuracy with higher rotation [26] (S3A and S3D Fig, S1 Table). We also reproduced the result regarding RT being shorter for familiar versus unfamiliar shapes [40] (S3B Fig, S1 Table), given that a more complex figure is more abstract to rotate spatially through visual imagery. Moreover, we reproduced results showing that RT are shorter for non-mirrored compared to mirrored figures [26] (S3C Fig, S1 Table), given that the mirrored image needs an extra rotation around the screen plane to confirm the response [25]. We also found an interaction between mirrored objects and the rotation angle (S1 Table), which is apparent in [41], as well, although the authors did not report it explicitly.

Finally, for the ANT, we reproduced the typical finding of longer RT for spatially neutral (vs. informative) cues (S4A Fig, S1 Table) and for conflict trials [27]. We also reproduced the interaction between cue and congruence conditions (S1 Table), that is, the effect of

incongruent flankers was stronger when cues contained irrelevant spatial information [27]. Results for accuracy could not be reproduced. Probably because of the scarcity of error events, this dependency may only be observed in larger samples.

## 4.2. Evaluation of test-retest reliability and internal consistency

Our test-retest and split-half ICC analyses demonstrated that RT in the considered tasks have adequate reliability for neuropsychological assessments, while, in general, accuracy, standard deviation of RT, and slopes of RT do not feature good reliability (Fig 2 and S5 Fig). ANT-related measures (i.e., alerting, orienting, and conflict) are moderately reliable. In line with [9], we recommend that the assessment of attention aspects should be preferentially made through the RT of correct trials, rather than accuracy or variability scores, as RT is usually found to be the most reliable measure [42]. Specifically, Steinborn and collaborators, evaluating a conceptually related test of concentration ability, stated that speed-based scores (i.e., based on RT) are a crucial and reliable dimension of attentional performance; that error scores should be used as a secondary measure (e.g. to check for aberrant behavior); and that variability scores should not be used at all [9]. However, it is often assumed that RT and errors are equipotent measures in psychometric assessments [43], despite the reported lack of reliability for error scores ([42], but see [44]. If one considers that the error scores are important in an assessment, a combination of RT and error scores is recommended to be used in the analysis [45–47], besides demonstrated psychometric properties. We also argue that poor ICCs might be due to the sparsity of errors and to RT being the natural primary measure of performance in speeded tasks, as is especially the case for the errors in CPT and ANT (as compared with ICC values for errors in Switcher and Mental Rotation tasks; see Fig 2). However, insufficient test–retest reliability of accuracy scores in attention-related tasks may also arise, at least partially, from violations of statistical assumptions due to the nature of the data. Since errors often exhibit a nonparametric distribution and floor effects, and the calculation of the ICC is subject to the assumption of normality and relies on stable interindividual differences (i.e., sufficient systematic variability), the estimated ICC for accuracy scores in attention-related tasks may suffer from the eventual violation of these assumptions [48–50].

In particular, the Switcher task's RT exhibits relatively lower test-retest reliability, as compared to the other tasks evaluated. As discussed earlier, the Switcher task also presents some issues with construct validity and, together with limited test-retest reliability, its application is compromised. Further studies are necessary to confirm whether a longer practice period may improve validity and reliability in this task. Low reliability can be improved, for example, by increasing the number of trials in a test [8] and, as the Switcher task in its current version is relatively short, we propose that future studies evaluate its test-retest reliability for different task lengths.

Both ICC and Pearson correlation have been used in previous psychometric evaluations to determine the degree of relationship in both test-retest and split-half analyses. Here, we used ICC instead of Pearson correlation to determine reliability because the ICC also considers the means and the variance to estimate the similarity between scores and not only the correlation [7, 8]. Reliability here can also be understood as the proportion of score variance explained by differences between subjects, with the remainder coming from a combination of random and systematic error [8].

## 4.3. Temporal stability

We observed that, in general, speed-related responses became faster when the task was administered a second time, revealing practice effects, or lack of temporal stability (Fig 3, Table 3).

However, it is important to mention that the presence of practice effects does not per se imply a lack of reliability, as shown above. A psychometric score can exhibit a strong sensitivity to practice and still demonstrate high test-retest reliability if practice-induced changes remain consistent (i.e., additive) across subjects over time [42]. For Study II, we acquired data from only two days of application, which is why we could only assess practice effects between these two days and the presumable stabilization of performance after even more practice is beyond the scope of this study. It is also worth noting that other measures, such as accuracy-, variability-, and performance-related scores did not exhibit practice effects but, as described earlier, lack test-retest reliability, compromising their use for individual assessment. Furthermore, practice effects were not observed for PVT RT, indicating that this task score is particularly suited for repeated assessments, for instance, when assessing alertness at different times of day [24, 51].

We also observed that, when only the second day of application is considered in the analysis of the Switcher task, the random switch condition leads to longer RT (as compared to a fixed–and therefore predictable–order of switching-feature rules) (S2A Fig, S2 Table), as would have been expected from face validity. A short practice period is present in its current PEBL implementation, but it was here demonstrated insufficient to lead to reliable switch cost differences. The lack of reliability might be because individuals need more time to adapt and normalize the influence of motor and visual reorienting (but not attentional switching) to the target before switching-related speed differences become measurable. We thus recommend that for measuring switch costs (i.e., RT differences between prepared and unprepared rule switching), a longer practice period in this task should be adopted. As the Switcher task is relatively short (duration of approximately 5 min), we recommend to at least double its application and consider the first application as a practice trial. Finally, while practice trials were not included for cognitively simple tasks (PVT) or tasks whose conditions were randomized (CPT and Mental Rotation task), they would be relevant for nonrandomized and intellectually more challenging tasks, such as the Switcher task.

### 4.4. Evaluation of stability across application sites

Applying the CPT and Switcher task at different sites (located in different countries) led to invariant RT results, indicating an insensitivity of these tasks to cultural differences (Fig 4) and experimental protocol differences (hardware, illumination, presentation of preliminary instructions). Along these lines, Conners and colleagues also observed that the CPT results were independent of the ethnicity of participants [18]. However, the application of the Switcher task at different sites led to different results for within-subject performance variability (i.e., standard deviation of RT; see S8 Table). Although we have already shown that variability measures did not achieve satisfactory test-retest reliability, this finding adds to the evidence for a lack of general reliability of the Switcher task. Further studies are necessary to show whether better reliability of the Switcher task is achieved after a longer practice period, which might improve the assessment of switch costs, as discussed above in the section 4.1.

### 4.5. Within- and between-task correlations

Presence and absence of correlations among task scores provide additional information about reliability and construct validity [3]. Specifically, the RT measures across conditions provided by the same task were in general highly correlated, which is evidence in favor of reliability/ internal consistency (as also indicated by the split-half ICC). This correlation would be expected because the within-task conditions are similar, e.g., speeded responses to visual stimuli. Other within-task correlations may indicate discriminant validity. For instance, the lack of correlation among ANT alerting, orienting, and conflict [27] is reproduced in our study and

speaks in favor of the (intended) orthogonality of the constructs. We also found positive between-task correlations, which may reflect that some attentional components are associated with speeded performance in more than one task, or that multiple attentional aspects are involved in the assessment–which indicates convergent validity. For instance, CPT RT is positively correlated with mental rotation latency in trials with highly rotated stimuli as well as with ANT RT scores (Fig 5). In this case, a common discriminative aspect of attention might be playing a similar role in all those scores (convergent validity), as subjects have to discriminate between two classes of stimuli in these tasks and either refrain from a motor response or provide an alternative response. In addition, alerting, as measured by the ANT, is moderately correlated with PVT scores (Fig 5), supporting the notion that the ability to control alertness is taxed by both tasks [27, 39]. An interesting result was that attentional orienting, also measured by the ANT, was negatively correlated with the PVT's RT slope and Switcher task scores (Fig 5 and S8 Fig), suggesting that effective spatial orienting of attention is also present in vigilance and task-switching abilities, measured by PVT and Switcher task, respectively (the higher the attentional orienting, the lower the vigilance detriment and the switching RT).

## 4.6. Associations with questionnaire scores

We found some relationships between task scores and questionnaires measuring attentional/motivational aspects, supporting construct validity through meaningful relationships with task performance. As for the DSSQ, we found that the higher the success motivation, the lower the RT slope in the PVT (Fig 6), reflecting better maintenance of alertness over time when participants report higher internal motivation to perform well in the assessment tasks. Moderate correlations between task and questionnaire scores, uncorrected for multiple comparisons, were convergent (S8 Fig). For example, we found that higher perceived intrinsic motivation was related to better performance in the PVT (overall RT and slope of RT) and more effective attentional orienting in the ANT, suggesting that this aspect of subjective task engagement is linked to a better capability of sustaining and orienting attention. Higher perceived task-related interference was associated with slower responses in both CPT (ISI2) and ANT (congruent condition), and higher task-irrelevant interference was also associated with slower responses in the ANT (congruent and incongruent conditions). These results indicate that intrusive thoughts that arise either from the task or independent of it scale with performance levels in only a subset of tasks. Intriguingly, higher levels of perceived control and confidence, as measured by the DSSQ, went along with slower responding in the ANT (congruent and incongruent conditions), indicating that participants reporting a stronger feeling of control took longer to respond in a task requiring the exertion of top-down control for correct response selection, perhaps reflecting interindividual differences in the speed–accuracy tradeoff.

Participants reporting higher levels of cognitive failures in everyday life according to the CFQ were found to show faster continuous discrimination in the CPT (ISI1), better mental rotation ability, and stronger performance deterioration (RT slope) in the ANT. This pattern may be related to the impulsivity associated with high CFQ scores, which may initially facilitate fast responding to unpredictable stimuli in a task but may also accelerate the deterioration of performance over time due to enhanced distractibility. As the ANT was the last task of the battery, ensuing fatigue may have augmented distractibility, which might explain why the relationship between performance deterioration and cognitive failure was only observed for this task.

The fact that task scores were not associated with the BRUMS scores suggests that performance of the evaluated tasks might not be substantially affected by the mood facets assessed by

this questionnaire. Taken together with the analogous absence of substantial relationships between task performance and the more affective sub-scores of the DSSQ, this insensitivity to a range of affective states might be considered a positive aspect of the tasks, which were not designed to detect differences in mood but attentional functions.

In general, however, expected associations between objective (cognitive tasks) and subjective (questionnaire) measures in psychometric evaluations often find only limited empirical support. Smit and colleagues (2021) summarized that the lack of convergence between subjective and objective instruments may result from various factors including measurement of different aspects of cognition; differences in motivation when performing tasks and completing questionnaires; successful subject-specific compensation during task performance; or a lack of ecological validity, sensitivity, or specificity of objective measures [52]. In our study, although such aspects may have influenced our results, we attribute the lack of convergence between objective and subjective measures primarily to our limited sample size and restricted variance. Therefore, caution is needed in interpreting these results, which might serve as initial evidence for an evaluation with a larger number of subjects. In our exploratory analysis, we were able to identify only strongly pronounced associations, whereas more subtle ones may have remained undetected.

## 4.7. Limitations

In this study, sample sizes were modest compared to other psychometric studies with similar aims. The tasks evaluated here were administered as part of fMRI-neurofeedback trials, which limited the acquisition of a larger sample size. In addition, assessment of the impact of the site of application was not possible for some tasks (i.e., PVT, Mental Rotation, and ANT) because either the application setting or the environmental context was not similar across sites. The low sample size also affected the applicability of the results to age groups (other than young adults) and left-handers, as well as prevented the examination of gender differences. While these aspects could be addressed in future large-scale efforts, this study constitutes initial evidence of psychometric properties of the tasks assessed. Due to the low sample size, our study was therefore only sensitive to large effects (Table 3 and S1, S2 and S8 Tables) and our initial assessment provides evidence for the most salient aspects of the tasks assessed, suggesting where adjustments might be needed in future task design and application. The low sample size also prevented the battery of tasks from being randomized, hence fatigue and other sequence effects might be also present in our evaluation and absolute levels of performance might have been affected; however, we attested to task-condition differences and validity and reliability of the tasks despite (rather than due to) order effects. Future research should evaluate how the psychometric characteristics observed here hold in larger and more diverse samples and how neurological and psychological conditions may impact the psychometric properties of these tasks. Therefore, the low sample size in this study, while limiting in several aspects, still yields valuable preliminary insights to be explored in a more comprehensive and definitive large-scale psychometric evaluation.

As the PEBL software is freely accessible, we should also acknowledge open-source concerns, broadly discussed in [11] in terms of limited modifiability of well-established tests [3]. One should keep in mind that the advantageous accessibility of PEBL to the whole scientific community might make it also susceptible to misapplications and individual task adaptations according to, for instance, pragmatic considerations. In the context of individual psychometric assessments, this should of course be avoided, and researchers aiming to assess individual differences with PEBL-based tasks should rely on task implementations whose psychometric properties have been properly evaluated.

## 5. Conclusions

Our study provides information about the reliability and validity of computerized tasks designed to assess various aspects of attention, as included in the PEBL software package. Scores reflecting individual performance speed were found to be reliable/internally consistent–despite general practice effects–, while scores reflecting accuracy, variability in performance speed, or time-related speed detriment were found to lack reliability. Moreover, we observed stability of CPT and Switcher task scores across sites. For all tasks evaluated, we corroborated internal consistency through within-task correlations. We also provided evidence for convergent validity observing common discriminative aspects of attention shared between tasks and discriminant validity through scores taxing different attentional aspects within the same task. Further evidence for construct validity was obtained through assessments of task-condition-specific effects and relationships between task and questionnaire scores. In its current PEBL implementation, the Switcher task appears to lack stability and construct validity. We argue that a longer practice period before the actual assessment might be necessary for achieving sound psychometric properties. Overall, our evaluation offers an initial basis for choosing specific computerized attentional assessments in future research and for critically interpreting results in basic and clinical research. However, further research is needed to obtain a more comprehensive picture of the psychometric quality of the tasks evaluated here, including the establishment of norms, and to provide a proper justification for calling these tasks attentional *tests*.

## Supporting information

**S1 Fig. Continuous Performance Task (CPT) scores across conditions.** CPT reaction time (RT) as a function of inter-stimulus interval (ISI) for single (A) and double applications (B). CPT standard deviation as a function of ISI for the single application (C). Asterisks represent significant differences in post-hoc tests corrected for multiple comparisons using the Sidak method (**** $p < 0.0001$, * $p < 0.05$). ISI1, ISI2, ISI4 = inter-stimulus interval of 1, 2, and 4 s, respectively.
(TIF)

**S2 Fig. Switcher task scores across conditions.** (A) Switcher task reaction time (RT) as a function of Type (Type1, Type2, Type3 = alternate switch, fixed switch, random switch, respectively) for double application. (B) Switcher task standard deviation as a function of Type for single application. Asterisks represent significant differences in post-hoc tests corrected for multiple comparisons using the Sidak method (** $p < 0.01$, * $p < 0.05$).
(TIF)

**S3 Fig. Mental Rotation task scores across conditions.** Mental Rotation task reaction time (RT) as a function of (A) rotation angle, (B) rotation angle and figure type (fig1 and fig2 = unfamiliar and familiar figures, respectively), and (C) on rotation angle and mirroring (diff and same represent mirrored and unmirrored figures, respectively). (D) Mental Rotation task accuracy (number of correct responses divided by total number of trials) as a function of rotation angle. Asterisks represent significant differences in post-hoc tests corrected for multiple comparisons using the Sidak method (**** $p < 0.0001$, *** $p < 0.001$, ** $p < 0.01$, * $p < 0.05$).
(TIF)

**S4 Fig. Attentional Network Test (ANT) scores across conditions.** ANT reaction time (RT) scores as a function of (A) cue type (cue1, cue2, cue3, and cue4 represent uncued, center cued,

top-bottom cued, and direction-cued trials, respectively) and (B) stimulus congruence. Asterisks represent significant differences in post-hoc tests corrected for multiple comparisons using the Sidak method ($^{****}$ p < 0.0001).
(TIF)

**S5 Fig. Split-half Intraclass Correlation Coefficients (ICC) for scores in a one-day application with 15 participants.** CPT = Continuous Performance Task; PVT = Psychomotor Vigilance Task; ANT = Attention Network Test; ISI1, ISI2, ISI4 = inter-stimulus interval of 1, 2, and 4 s, respectively; Type1, Type2, and Type3 = alternate switch, fixed switch, and random switch, respectively, CB = confidence boundary. Blue, green, yellow, and orange colors indicate excellent (ICC $\geq$ 0.9), good (0.75 $\leq$ ICC < 0.9), moderate (0.5 $\leq$ ICC < 0.75), and poor (ICC < 0.5) ICC values.
(TIF)

**S6 Fig. Attentional Network Test (ANT) scores across days.** (A) ANT RT scores as a function of stimulus congruence and measurement session (day). (B) Attentional Network Test (ANT) conflict score as a function of measurement session. Day 1 = first application, Day 2 = repeated application (i.e., after practice). Asterisks represent significant differences in post-hoc tests corrected for multiple comparisons using the Sidak method ($^{****}$ p < 0.0001, $^{**}$ p < 0.01).
(TIF)

**S7 Fig. Correlation among task scores on the second day of application.** Significant positive and negative correlations (FDR-corr. p < 0.05) are reported in green and red, respectively. Correlations with p < 0.05, uncorrected for multiple comparisons, are printed in bold. CPT = Continuous Performance Task; PVT = Psychomotor Vigilance Task; ANT = Attention Network Test; RT = reaction time; ISI1, ISI2, ISI4 = inter-stimulus interval of 1, 2, and 4 s, respectively; type1, type2, type3 = alternate switch, fixed switch, random switch, respectively.
(TIF)

**S8 Fig. Significant correlations uncorrected for multiple comparisons (unc. p < 0.05).** CPT = Continuous Performance Task; RT = reaction time; ISI1 and ISI2 = inter-stimulus interval of 1 and 2 s, respectively; DSSQ = Dundee Stress State Questionnaire; PVT = Psychometric Vigilance Task; ANT = Attentional Network Test; CFQ = Cognitive Failures Questionnaire.
(TIF)

**S1 Table. Statistics of significant main effects and interactions across conditions within tasks and post-hoc analyses.**
(DOCX)

**S2 Table. Statistics of significant main effects and interactions and post-hoc analyses for double application (Study II).**
(DOCX)

**S3 Table. Descriptive statistics for performance measures of the CPT (Continuous Performance Task).**
(DOCX)

**S4 Table. Descriptive statistics for performance measures of the Switcher Task.**
(DOCX)

**S5 Table. Descriptive statistics for performance measures of the PVT (Psychomotor Vigilance Task).**
(DOCX)

**S6 Table. Descriptive statistics for performance measures of the Mental Rotation task.**
(DOCX)

**S7 Table. Descriptive statistics for performance measures of the Attentional Network Test.**
(DOCX)

**S8 Table. Statistics of significant main effects and interactions across sites and post-hoc analyses.**
(DOCX)

**S9 Table. Significant correlations between task scores and questionnaire scores.**
(DOCX)

## Author Contributions

**Conceptualization:** Robert Langner, Brian J. Piper, Gustavo S. P. Pamplona.

**Data curation:** Gustavo S. P. Pamplona.

**Formal analysis:** Gustavo S. P. Pamplona.

**Funding acquisition:** Frank Scharnowski, Silvio Ionta, Carlos E. G. Salmon, Gustavo S. P. Pamplona.

**Investigation:** Gustavo S. P. Pamplona.

**Methodology:** Robert Langner, Frank Scharnowski, Carlos E. G. Salmon, Gustavo S. P. Pamplona.

**Project administration:** Frank Scharnowski, Carlos E. G. Salmon.

**Resources:** Frank Scharnowski, Carlos E. G. Salmon.

**Software:** Brian J. Piper.

**Supervision:** Frank Scharnowski, Silvio Ionta, Carlos E. G. Salmon.

**Validation:** Gustavo S. P. Pamplona.

**Visualization:** Gustavo S. P. Pamplona.

**Writing – original draft:** Gustavo S. P. Pamplona.

**Writing – review & editing:** Robert Langner, Frank Scharnowski, Silvio Ionta, Carlos E. G. Salmon, Brian J. Piper, Gustavo S. P. Pamplona.

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
