## [Decision Letter · Decision Letter 0]

7 Sep 2022

PONE-D-22-16441Evaluation of the reliability and validity of computerized tests of attentionPLOS ONE

Dear Dr. Pamplona,

Thank you for submitting your manuscript to PLOS ONE. After careful consideration, we feel that it has merit but does not fully meet PLOS ONE’s publication criteria as it currently stands. Therefore, we invite you to submit a revised version of the manuscript that addresses the points raised during the review process.

Please note that we have only been able to secure a single reviewer to assess your manuscript. We are issuing a decision on your manuscript at this point to prevent further delays in the evaluation of your manuscript. Please be aware that the editor who handles your revised manuscript might find it necessary to invite additional reviewers to assess this work once the revised manuscript is submitted. However, we will aim to proceed on the basis of this single review if possible. The reviewer has identified a number of concerns that need to be carefully addressed in a revision of the manuscript. Please respond to all of the reviewer's comments, paying particular attention to their requests for methodological clarifications and their suggestions for improvements to the contextualisation and discussion of your findings.

We look forward to receiving your revised manuscript.

Kind regards,

Jamie Males

Editorial Office

PLOS ONE

Journal Requirements:

“This work was supported by the Swiss National Science Foundation (grants PZ00P1_170506/1, PP00P1_202665/1, BSSG10_155915, 100014_178841, 32003B_166566, and PP00P1_170506/1), by the Brazilian National Council for Scientific and Technological Development (CNPq), the Brazilian National Council for the Improvement of Higher Education (CAPES), the Foundation for Research in Science and the Humanities at the University of Zurich (STWF-17-012), the Baugarten Stiftung, and the Swiss Government Excellence Scholarship.”

“F.S. was supported by the Swiss National Science Foundation (grants BSSG10_155915, 100014_178841, 32003B_166566, and PP00P1_170506/1), the Foundation for Research in Science and the Humanities at the University of Zurich (STWF-17-012), and the Baugarten Stiftung. S.I. was supported by the Swiss National Science Foundation (grants PZ00P1_170506/1 and PP00P1_202665/1). G.S.P.P was supported by the Brazilian National Council for Scientific and Technological Development (CNPq), the Brazilian National Council for the Improvement of Higher Education (CAPES) and the Swiss Government Excellence Scholarship. The funders had no role in study design, data collection and analysis, decision to publish, or preparation of the manuscript.”

Reviewers' comments:

Reviewer's Responses to Questions

**Comments to the Author**

1. Is the manuscript technically sound, and do the data support the conclusions?

Reviewer #1: Yes

2. Has the statistical analysis been performed appropriately and rigorously? 

Reviewer #1: Yes

3. Have the authors made all data underlying the findings in their manuscript fully available?

Reviewer #1: Yes

4. Is the manuscript presented in an intelligible fashion and written in standard English?

Reviewer #1: Yes

5. Review Comments to the Author

Reviewer #1: The study aims to measure the psychometric properties of five attention tests developed in free access software with the aim that this software can be used and has sufficient reliability. The authors find better psychometric properties for the response time (RT) measures collected in the five tests than for the precision measures (errors). Likewise, some tests obtain better reliability and validity results than others, which suggests that these tests would be the appropriate ones for their possible use. The study is relevant to the extent that the analyzes carried out are necessary to be able to safely use these novel attention tests. The study determines which tests should be used and which should not.

Although true, the study has some limitations. The sample is not large and has been collected in three different studies. The authors make this, in part, an advantage by analyzing the differences between places, but the truth is that the methodological differences of the three subsamples limit the quality of the sample. Likewise, sometimes it is ambitious to present certain analyzes such as the comparison between sites, the correspondence with self-report questionnaires, etc. when later it is not deepened much into it.

Here are some recommendations for authors:

METHODS

1. I recommend that authors review the formatting of tables to conform to APA style.

2. The introduction indicates that five tests were applied. In the "application procedure" section, it is indicated again that these same tests were applied in studies I and II (The sequence for Studies I and II was CPT, Switcher, PVT, Mental Rotation, and ANT). No clutch indicates that the Stroop test was applied instead of the ANT test (Study III it was CPT, Switcher, PVT, and Stroop. ). The stroop test is not mentioned again in the article. I suggest the authors clarify this discrepancy.

3. I recommend that the authors include in the procedure how long it took to apply the five tests on average.

4. As it is a sample collected through three different studies, they have collected the associations with questionnaire scores with only a third of the sample, which significantly reduces the number of subjects analyzed in this type of analysis (convergent validity).

5. Lines 191-192: The following citation is missing the parentheses in the year: Anderson, Deane, Lindley, Loucks, & Veach, 2012.

Line 529: The following citation is missing the parentheses in the year: Berteau-Pavy et al., 2011,

Line 694: The following citation is missing the parentheses in the year: Mueller & Piper, 2014

6. I recommend that the authors include information on the psychometric properties of the self-report questionnaires they have used, for example Crombach's alpha

DISCUSSION

7. I encourage the authors to expand the discussion on the lack of correlation between some objective tests and self-report questionnaires, considering that the sense of efficacy perceived is subjective and, in many cases, does not coincide with that performed.

I suggest the authors read articles that address this aspect, such as:

• Smit, D., Koerts, J., Bangma, D. F., Fuermaier, A. B., Tucha, L., & Tucha, O. (2021). Look who is complaining: Psychological factors predicting subjective cognitive complaints in a large community sample of older adults. Applied Neuropsychology: Adult, 1-15.

8. I also suggest the authors to discuss in more depth not having found test-retest reliability in accuracy measures. In this sense, I recommend considering aspects that are known to affect the reliability of commission errors, for example the absence of normality and ground effect in some of the measurements.

I suggest the authors read articles that address this aspect, such as:

• Fernández-Marcos, T., de la Fuente, C., & Santacreu, J. (2018). Test–retest reliability and convergent validity of attention measures. Applied Neuropsychology: Adult, 25(5), 464-472.

• Shaked, D., Faulkner, L. M., Tolle, K., Wendell, C. R., Waldstein, S. R., & Spencer, R. J. (2020). Reliability and validity of the Conners’ continuous performance test. Applied Neuropsychology: Adult, 27(5), 478-487.

• Wilding, J., & Cornish, K. (2007). Independence of speed and accuracy in visual search: Evidence for separate mechanisms. Child Neuropsychology, 13(6), 510–521. doi:10.1080/ 09297040601160574

6. PLOS authors have the option to publish the peer review history of their article (what does this mean?). If published, this will include your full peer review and any attached files.

Reviewer #1: **Yes: **Tatiana Fernández-Marcos

---

## [Author Response · Author response to Decision Letter 0]

17 Oct 2022

Response to Reviews

1. Reviewer 1

The study aims to measure the psychometric properties of five attention tests developed in free access software with the aim that this software can be used and has sufficient reliability. The authors find better psychometric properties for the response time (RT) measures collected in the five tests than for the precision measures (errors). Likewise, some tests obtain better reliability and validity results than others, which suggests that these tests would be the appropriate ones for their possible use. The study is relevant to the extent that the analyzes carried out are necessary to be able to safely use these novel attention tests. The study determines which tests should be used and which should not.

Although true, the study has some limitations. The sample is not large and has been collected in three different studies. The authors make this, in part, an advantage by analyzing the differences between places, but the truth is that the methodological differences of the three subsamples limit the quality of the sample. Likewise, sometimes it is ambitious to present certain analyzes such as the comparison between sites, the correspondence with self-report questionnaires, etc. when later it is not deepened much into it.

Here are some recommendations for authors:

1.1. Comment 1

I recommend that authors review the formatting of tables to conform to APA style.

We appreciate the observation. All tables in the main text and supplementary material were formatted according to the APA style. 

1.2. Comment 2

The introduction indicates that five tests were applied. In the "application procedure" section, it is indicated again that these same tests were applied in studies I and II (The sequence for Studies I and II was CPT, Switcher, PVT, Mental Rotation, and ANT). No clutch indicates that the Stroop test was applied instead of the ANT test (Study III it was CPT, Switcher, PVT, and Stroop. ). The stroop test is not mentioned again in the article. I suggest the authors clarify this discrepancy.

We edited the section to clarify how the task batteries were comprised and the discrepancies between them. We also clarified the implication of the discrepancies in terms of further assessment (line 148):

“Studies I and II comprised a battery of five tasks: CPT, Switcher, PVT, Mental Rotation, and ANT; Study III consisted of four tasks: CPT, Switcher, PVT, and Stroop. Therefore, CPT, Switcher, and PVT were common to all studies (and applied in the same order), but Studies I and II included Mental Rotation and ANT and Study III included the Stroop task. As our evaluation of reliability relies on repeated assessments and stability across sites, and the Stroop Task was applied at only one site in a single application, this task was not further considered here. Furthermore, since Mental Rotation and ANT were applied at only one site (Studies I and II), these tasks were not considered for the assessment of stability across sites.”

1.3. Comment 3

I recommend that the authors include in the procedure how long it took to apply the five tests on average.

Following the suggestion of the reviewer, we are now including study-specific information about application duration in line 189:

“The total duration of the application, including breaks, was 1h25min ± 9min for Study I, 1h21min ± 3min for the first day of Study II, 1h18min ± 5 min for the second day of Study II, and 56min ± 6min for Study III (the latter study had one task less and one break less, as compared to Studies I and II).”

1.4. Comment 4

As it is a sample collected through three different studies, they have collected the associations with questionnaire scores with only a third of the sample, which significantly reduces the number of subjects analyzed in this type of analysis (convergent validity).

Thanks for raising this issue. Please note, however, that the associations between questionnaire scores and task measures for two of the questionnaires (DSSQ and CFQ) were based on two-thirds of the sample, since these questionnaires were applied in Studies I and II. As the reviewer pointed out, the association with BRUMS scores could only be assessed for approximately one-third of the sample, since this questionnaire was applied only in Study III. Associations with BRUMS scores were not statistically significant. To clarify which associations between questionnaires and study-specific measures were evaluated, we included the following sentence in the main text (line 375):

“Following the questionnaires applied in each study, we evaluated the associations between DSSQ and CFQ scores with the task measures from Studies I + II, and the associations between BRUMS scores and the task measures from Study III.”

We reported in the text that only one correlation between questionnaire scores and task measures survived the correction for multiple comparisons (lines 486). Hence, it is unlikely that the reported result is a false positive. Other correlations, uncorrected for multiple comparisons, were also reported for completeness (lines 489). We have included the following sentence to indicate that the reader should be cautious about interpretating the results (line 706):

“In our study, although such aspects may have influenced our results, we attribute the lack of convergence between objective and subjective measures primarily to our limited sample size and restricted variance. Therefore, caution is needed in interpreting these results, which might serve as initial evidence for an evaluation with a larger number of subjects. In our exploratory analysis, we were able to identify only strongly pronounced associations, whereas more subtle ones may have remained undetected.”

Finally, we would like to point out that all significant correlations (corrected and uncorrected for multiple comparisons, Figs. 6 and S8, respectively) were reported as scatterplots. Their patterns indicate that the results were not driven by outliers, which is a general concern when reporting correlations, especially in samples of modest size like in our case.

1.5. Comment 5

Lines 191-192: The following citation is missing the parentheses in the year: Anderson, Deane, Lindley, Loucks, & Veach, 2012.

Line 529: The following citation is missing the parentheses in the year: Berteau-Pavy et al., 2011,

Line 694: The following citation is missing the parentheses in the year: Mueller & Piper, 2014

We thank the reviewer for the attention to the details. Issues with formatting the citation arose because of the used reference manager. We reviewed the manuscript thoroughly and corrected the mentioned citations, as well as other ones, to the Vancouver style, the one required by PlosOne. 

1.6. Comment 6

I recommend that the authors include information on the psychometric properties of the self-report questionnaires they have used, for example Crombach's alpha

We thank the reviewer for this recommendation. According to previous studies on the psychometric properties of the questionnaires used, they present good reliability based on Cronbach’s alpha. We have now included this information in the manuscript, as suggested by the reviewer (line 159): 

“All the questionnaires have been reported to have acceptable to very good reliability: Cronbach’s alpha values were found to be in the range of 0.76–0.89 for DSSQ (25), 0.92–0.93 for CFQ (28), and 0.76–0.85 for BRUMS (29).”

1.7. Comment 7

DISCUSSION

I encourage the authors to expand the discussion on the lack of correlation between some objective tests and self-report questionnaires, considering that the sense of efficacy perceived is subjective and, in many cases, does not coincide with that performed.

I suggest the authors read articles that address this aspect, such as:

• Smit, D., Koerts, J., Bangma, D. F., Fuermaier, A. B., Tucha, L., & Tucha, O. (2021). Look who is complaining: Psychological factors predicting subjective cognitive complaints in a large community sample of older adults. Applied Neuropsychology: Adult, 1-15.

We agree with the reviewer. The lack of convergence between objective and subjective measurements is a well-known and long-standing problem in psychology. We appreciate the article recommendation, which brings a comprehensive summary of possible reasons for this lack of convergence. In addition, we also attribute this issue in our study to limited sample size and variance, which may have restricted the findings to only those with large effect sizes. We have included the following excerpt in the discussion (line 700):

“In general, however, expected associations between objective (psychological tasks) and subjective (questionnaire) measures in psychometric evaluations often find only limited empirical support. Smit and colleagues (2021) summarized that the lack of convergence between subjective and objective instruments may result from various factors including measurement of different aspects of cognition; differences in motivation when performing tasks and completing questionnaires; successful subject-specific compensation during task performance; or a lack of ecological validity, sensitivity, or specificity of objective measures (Smit et al., 2021). In our study, although such aspects may have influenced our results, we attribute the lack of convergence between objective and subjective measures primarily to our limited sample size and restricted variance. Therefore, caution is needed in interpreting these results, which might serve as initial evidence for an evaluation with a larger number of subjects. In our exploratory analysis, we were able to identify only strongly pronounced associations, whereas more subtle ones may have remained undetected.”

1.8. Comment 8

I also suggest the authors to discuss in more depth not having found test-retest reliability in accuracy measures. In this sense, I recommend considering aspects that are known to affect the reliability of commission errors, for example the absence of normality and ground effect in some of the measurements.

I suggest the authors read articles that address this aspect, such as:

• Fernández-Marcos, T., de la Fuente, C., & Santacreu, J. (2018). Test–retest reliability and convergent validity of attention measures. Applied Neuropsychology: Adult, 25(5), 464-472.

• Shaked, D., Faulkner, L. M., Tolle, K., Wendell, C. R., Waldstein, S. R., & Spencer, R. J. (2020). Reliability and validity of the Conners’ continuous performance test. Applied Neuropsychology: Adult, 27(5), 478-487.

• Wilding, J., & Cornish, K. (2007). Independence of speed and accuracy in visual search: Evidence for separate mechanisms. Child Neuropsychology, 13(6), 510–521. doi:10.1080/ 09297040601160574

The reviewer raised an important point and we complemented the discussion based on her comment (lines 577ff.): 

“However, insufficient test–retest reliability of accuracy scores in attention-related tasks may also arise, at least partially, from violations of statistical assumptions due to the nature of the data. Since errors often exhibit a nonparametric distribution and floor effects, and the calculation of the ICC is subject to the assumption of normality and relies on stable interindividual differences (i.e., sufficient systematic variability), the estimated ICC for accuracy scores in attention-related tasks may suffer from the eventual violation of these assumptions [42–44].”

---

## [Decision Letter · Decision Letter 1]

2 Jan 2023

PONE-D-22-16441R1Evaluation of the reliability and validity of computerized tests of attentionPLOS ONE

Dear Dr. Pamplona,

Thank you for submitting your manuscript to PLOS ONE. After careful consideration, we feel that it has merit but does not fully meet PLOS ONE’s publication criteria as it currently stands. Therefore, we invite you to submit a revised version of the manuscript that addresses the points raised during the review process.

We look forward to receiving your revised manuscript.

Kind regards,

Gabriel G. De La Torre

Academic Editor

PLOS ONE

Journal Requirements:

Reviewers' comments:

Reviewer's Responses to Questions

**Comments to the Author**

1. If the authors have adequately addressed your comments raised in a previous round of review and you feel that this manuscript is now acceptable for publication, you may indicate that here to bypass the “Comments to the Author” section, enter your conflict of interest statement in the “Confidential to Editor” section, and submit your "Accept" recommendation.

Reviewer #1: All comments have been addressed

Reviewer #2: (No Response)

2. Is the manuscript technically sound, and do the data support the conclusions?

Reviewer #1: Yes

Reviewer #2: Yes

3. Has the statistical analysis been performed appropriately and rigorously? 

Reviewer #1: Yes

Reviewer #2: Yes

4. Have the authors made all data underlying the findings in their manuscript fully available?

Reviewer #1: Yes

Reviewer #2: Yes

5. Is the manuscript presented in an intelligible fashion and written in standard English?

Reviewer #1: Yes

Reviewer #2: Yes

6. Review Comments to the Author

Reviewer #1: (No Response)

Reviewer #2: The authors present a study of the validity and reliability of 5 widely used computerized neuropsychological tests included in an open source software program (PEBL ,Psychology Experiment Building Language).

While this is an interesting tool and computerized neuropsychological assessment is gaining popularity since combining computerized tools and standard neuropsychological tests seems to provide the greatest value in clinical assessment, the sample size is an important limitation to draw firm conclusions but still the authors made a useful starting point.

Specific considerations are listed below:

Comment 1: In the second paragraph in the ‘Introduction’ section, I suggest to briefly mention the most appropriate statistical methods used to measure reliability and validity constructs.

Comment 2: In line 71 the limited evidence of the PEBL should be described in depth with the references and limitations of these studies.

Comment 3: In line 73 ‘Among other mental functions, PEBL enables the assessment of various…’ I recommend to better use cognitive functions instead of mental functions.

Comment 4: The description of the tasks (line 74 to 92) should be included in the Methods Section (Experimental procedures)

Comment 5: I suggest the authors to add to the section Experimental procedures ‘and measures’. For improving the value of this study, I suggest organizing this section:

- First part, paragraph 1 (line 139 to 147)

- Second part, paragraph 3 (line 168 to 186)

- Third part would include the procedure (a graphical explanation of the study design would help) and the description of the tasks including the measures that were selected for this study (Table 2).

Comment 6: The section Data analysis includes repeated information (line 279 to 281).

Comment 7: Although the PEBL (and e.g. in line 700 in this study) uses the term psychological tasks to refer to the CPT, Digits Span and so on, I would consider using a more specific term such as attentional, cognitive or neuropsychological tasks since it could be confusing.

7. PLOS authors have the option to publish the peer review history of their article (what does this mean?). If published, this will include your full peer review and any attached files.

Reviewer #1: **Yes: **Tatiana Fernández-Marcos

Reviewer #2: **Yes: **Manuela Martin-Bejarano Garcia

---

## [Author Response · Author response to Decision Letter 1]

17 Jan 2023

The manuscript was reviewed according to the reviewer's comments. Please refer to "Response to reviewers II"

---

## [Editor Report · Decision Letter 2]

18 Jan 2023

Evaluation of the reliability and validity of computerized tests of attention

PONE-D-22-16441R2

Dear Dr.Pamplona,

We’re pleased to inform you that your manuscript has been judged scientifically suitable for publication and will be formally accepted for publication once it meets all outstanding technical requirements.

Kind regards,

Gabriel G. De La Torre

Academic Editor

PLOS ONE